# Quantitative analysis of trabecular bone tissue cryosections via a fully automated neural network-based approach

Christopher Pohl[1]*, Moritz Kunzmann[2], Nico Brandt[1], Charlotte Koppe[1], Janine Waletzko-Hellwig[3,5], Rainer Bader[5], Friederike Kalle[4], Stephan Kersting[1], Daniel Behrendt[1], Michael Schlosser[1†], Andreas Hoene[1]

1 Department of General Surgery, Visceral, Thoracic and Vascular Surgery, University Medical Center Greifswald, Greifswald, Germany, 2 University of Heidelberg, BioQuant Center, Heidelberg, Germany, 3 Department of Oral, Maxillofacial and Plastic Surgery, Rostock University Medical Center, Rostock, Germany, 4 Department of Oto-Rhino-Laryngology, Head and Neck Surgery, Rostock University Medical Center, Rostock, Germany, 5 Department of Orthopaedics, Research Laboratory of Biomechanics and Implant Technology, Rostock University Medical Center, Rostock, Germany

† Deceased.

* christopher.pohl@uni-greifswald.de

**Data Availability Statement:** We uploaded the used code, installation instructions and all required information to reproduce created results to a github repository. https://github.com/ChrisPohl/

## Abstract

Cryosectioning is known as a common and well-established histological method, due to its easy accessibility, speed, and cost efficiency. However, the creation of bone cryosections is especially difficult. In this study, a cryosectioning protocol for trabecular bone that offers a relatively cheap and undemanding alternative to paraffin or resin embedded sectioning was developed. Sections are stainable with common histological dying methods while maintaining sufficient quality to answer a variety of scientific questions. Furthermore, this study introduces an automated protocol for analysing such sections, enabling users to rapidly access a wide range of different stainings. Therefore, an automated 'QuPath' neural network-based image analysis protocol for histochemical analysis of trabecular bone samples was established, and compared to other automated approaches as well as manual analysis regarding scattering, quality, and reliability. This highly automated protocol can handle enormous amounts of image data with no significant differences in its results when compared with a manual method. Even though this method was applied specifically for bone tissue, it works for a wide variety of different tissues and scientific questions.

## Introduction

Histological imaging has long played an important role in tissue analysis. It allows for qualitative statements on tissue and cells through different stains to highlight a multitude of different targets capable of answering countless research questions. However, for hard tissues including trabecular bone, the creation of histological sections is especially difficult. The compact spongy structure usually requires prior decalcification of the tissue to ensure viable sectioning, otherwise the tissue crumbles under the blade. The commonly used decalcification methods are

Bone-cryosection-segmentation Our image and created classifier data are available at Zenodo, published under the doi: 10.5281/zenodo.10560703.

**Funding:** This work was funded by the ESRE, ESF and ELER as well as the Ministry of education, science, and culture of Mecklenburg Western Pomerania 'Exzellenzforschungsprogramm' and is part of the HOGEMA joint research project under the sponsorship contract ESF/14-BM-A55-0015/18. The funders had no role in study design, data collection and analysis, decision to publish, or preparation of the manuscript.

**Competing interests:** The authors have declared that no competing interests exist.

either based on strong acids that can be detrimental to the tissue and further stainability but are very time efficient, or are based on ethylenediaminetetraacetic acid (EDTA) which is a gentler approach, but takes more time to fully decalcify a sample. With special approaches, like resin based embedding media or a special adhesive foil, decalcification can be avoided, but these methods are either highly technically demanding, expensive and/or inaccessible for most laboratories [1–3]. Following the decalcification process, embedding plays an important role for sectioning. The most common method for bone tissue is paraffin-based embedding, which has been in use for decades [4] as well as polymer-based methods [5–7] that were later optimized by using methyl methacrylate as embedding media [2, 7]. However, these methods come with a detriment. In methacrylate embedded sections antigens of the original sample are masked, negatively affecting their detection by specific detector antibodies utilized in immunohistochemical characterization. Countermeasures, if available, require technically demanding workarounds [8]. The time efficient and cheap cryosectioning, is a valid and often used method for soft tissues that requires no further workflow steps prior to immunohistochemical staining since antigens of the tissue are unmasked. But due to the complicated behaviour of hard tissues, cryosectioning is still a seldom applied approach in this field of research.

Histological staining is most often utilized to show qualitative effects; however, it also allows for numerous quantitative analyses. Histological quantitative results always need a multitude of samples, and therefore the interconnected manual workload was always the limiting factor. However, newer automated methods reduce the limitations of quantitative histology. Quantitative analysis of histological imaging is commercially available through digital analysis software like ZEISS ZEN—Digital Imaging for light-microscopy (Carl Zeiss MicroImaging GmbH, Jena, Germany). It is reliable, accurate and user-friendly, but is quite expensive to acquire and limited in their accessibility as it is mostly licensed or hardware locked to one workstation in the institution, if available at all [9, 10]. Well and often used tools for image analysis are freely available software like ImageJ [11] for its fast, easy and reproducible image quantification [12, 13]. They are able to process tremendous amounts of high-resolution datasets in a fraction of the time that a manual analysis would be possible of handling. Even though most of those tools are modular and offer possibilities for automation of processes, a lot of research groups do not utilize them to their full potential and stick to several different manual protocols and plugins, that exist for grid-supported cell counting, tissue area calculation and other analysis approaches. Though these processes are valid cell counting methods, they are still manual techniques, limited to fewer numbers of images.

However, with the development towards datasets of ever-increasing size, automation becomes more important for scientific research than ever, and neural network-based analysis provides the potential to improve quantitative image analysis even further. Neural networks are powerful methods, which approximate functions and dynamics by learning from examples [14]. They are able to include data amounts just as huge as other classical automated approaches, while maintaining the same precision of qualified investigators, even allowing for otherwise highly time intensive or impossible applications. In recent years, neural network-based image analysis is a quickly evolving field that gained significance in multiple diagnostic fields [15]. But it also started to gain importance in medical research [16–19]. Lately neural network-based analysis became more accessible with the release of freely available analysis software with built in network functionality. Today a multitude of tools, plugins, repositories and web applications allow for relatively easy access to network-based analysis [20]. For this protocol we focused on a QuPath-based approach (The University of Edinburgh, Scotland, Version 0.4.3; [21]. It offers basic neural-network based segmentation, while remaining easy to use and applicable for multiple scientific questions. Built in tools of these softwares allow for

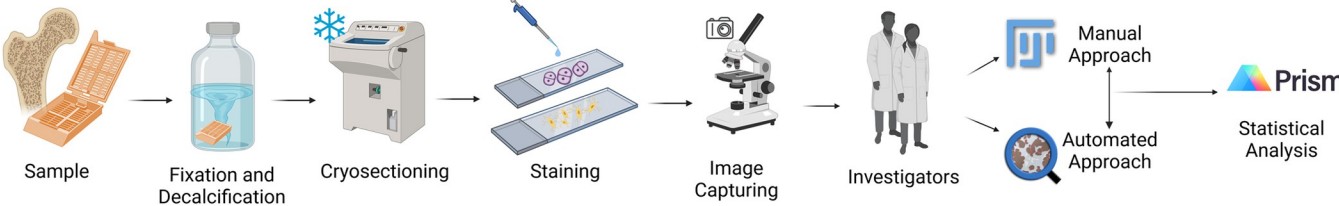

**Fig 1. Methodical overview of performed workflow.**

easy combination of multiple workflows, like simultaneous automated cell counting, network-based classification and multi-image processing.

In consequence, this study puts effort into establishing a cryosectioning method of decalcified bone tissue for subsequent quantitative histological analyses via automated workflows, that is adaptable for numerous scientific questions.

## Materials and methods

A methodical overview of various steps from initial sample preparation, morphometrical analysis to statistical data assessment is given in (Fig 1).

### Tissue sample origin

Bone samples were extracted post mortem from tissue donors in the Institute of Anatomy, Rostock University Medical Centre, under the approval of the Ethical Committee of the Medical Faculty of the University of Rostock (No. A 2016–0083) from late 2019 until early 2020. Samples were either cut out as bone platelets or cylindrical bone punches. Both sample types originate from distal human femoral condyles at sizes differing between 1–4 cm$^3$. Following the extraction, bone tissue samples were kept in phosphate buffered saline (PBS) (PanBiotech, Aidenbach, Germany).

### Tissue preparation

Prior to decalcification of bone platelets, a fixation step was performed to ensure cell integrity during decalcification. Therefore, 60 mM/L $KH_2PO_4$ (Sulpeco, Bellefonte, PA, USA) was combined with 80 mM/L $Na_2HPO_4$ (VWR, Darmstadt, Germany) and diluted in 860 mL aqua dest. Finally, 140 mL formalin (wt.37%; Sigma Aldrich, St. Louis, MO, USA) was added to the solution (VWR, Darmstadt, Germany). The samples were exposed to the fixation medium for 24 h in 150 mL solution per sample at room temperature. After fixation, samples were rinsed for two hours with tap water.

Because of the spongy, brittle texture of bone, tissue decalcification is required to ensure tissue integrity during sectioning. Therefore, an EDTA decalcification method known for its low impact on cell vitality and staining was applied [22]. 14 g EDTA (Sigma Aldrich, St. Louis, Missouri, USA) were combined with 9 mL ammonia solution (Merck, Darmstadt, Germany) and 76 mL aqua dest. to form the final decalcification solution used in this work. Other commercially available EDTA decalcification solutions would also be suitable. Bone tissue was kept in constant motion via a stirring magnet to reduce saturation effects of the solution. Whenever the solution reached a high turbidity, bone platelets were transferred into new medium. Usually after 96h for the first time, afterwards in greater intervals. Total decalcification time differed from seven days to six weeks, heavily depending on the size of the bone plate. To verify sufficient decalcification, a needle test was performed. It is important to check the core of the

platelets, if the needle reaches the core without creating cracks nor crunching noises, the tissue is ready for further processing.

For sectioning, tissue samples were placed in an embedding medium consisting of a 1/1 ratio of TissueTek (Sakura, Osaka, Japan) and PBS (PanBiotech, Aidenbach, Germany) and shock frozen in liquid nitrogen in square Peel-A-Way containers (TED- PELLA, INC, Redding, Canada). The dilution step enables the TissueTek solution to reach and fill all cavities of the bone tissue, while still avoiding freezing associated damages to the tissue. It is helpful to move the sample in the embedding medium until no further air bubbles emerge from it. Air bubbles create differences in resistance during sectioning and can therefore cause ruptures.

## Histology

Cryosections (8 to 10 μm thickness) of decalcified bone tissue were sectioned with a Leica CM 3050 S Cryotome (Leica, Nussloch, Germany). Finding a proper speed for sectioning is important. A too slow sectioning speed may result in compression of the tissue, while a fast speed may tear and rip the bone tissue apart. If sectioning proves to be very problematic, a 90˚ turn of the sample on the sled may help to secure sufficient results. If the sample is of cubic shape we recommend to align the blade with the tissue so that the initial contact-area of blade to tissue is maximised.

HE staining was performed according to standard protocols [23, 24]. Sections were fixated in ice-cold acetone and dried for at least 30 minutes at 37˚C. Afterwards the slides were put into PBS for five minutes followed by two minutes in aqua dest. The slides were then stained with filtered Mayer's Haematoxylin Solution (Sigma Aldrich, St. Louis, Missouri, USA) for either 10 or 15 minutes followed by washing with flowing tap water for four minutes until nuclei are stained blue. Counterstaining was performed with aqueous Eosin Y solution (Sigma Aldrich, St. Louis, Missouri USA) for two minutes, followed by another washing step in tap water for four minutes. After drying, slices were analysed with a light microscope CX41 (Olympus, Hamburg, Germany) combined with a digital colour camera DP20 (1600 × 1200 Pixel, Olympus, Hamburg, Germany).

For fluorescence staining, sections were fixed in 4% paraformaldehyde for 10 minutes at room temperature to preserve the cellular structure. Following fixation, the sections were stained using an *in situ* cell death detection kit (Roche, Basel, Switzerland) according to manufacturer's instructions. To aid in visualizing tissue architecture, stained sections were counterstained with DAPI (Molecular Probes, Eugene, OR, United States).

## Manual image analysis

Manual analysis of the image data followed a standard protocol to detect cells and tissue area previously published by our working group. (Walschus et al. 2011). Three independent investigators annotated the tissue area and counted the cells by hand. They used the ImageJ software (National Institutes of Health, USA, Version: 1.52a) and their built in plugins: "Grid", which creates grids on the pictures for better localisation and "Cell counter", which keeps track of counted cells. For validation of the other analysis methods regarding accuracy, variance, and quality, we used this well-established method for comparison.

## Automated tissue detection

We chose 3 different automated methods for tissue detection. Classical Otsu Thresholding [25], U-Net [26], a convolutional neuronal network and a QuPath-based neural network for pixel classification [21]. Otsu thresholding is available as a built in function of many software tools. For simplicity, we chose a setup that was performed in a Google Colab environment

using Python and the scikit-image library [27]. Both Networks, QuPath and U-Net, were equally trained with 20 image mask ground truths of tissue area, annotated in QuPath by three different trained Investigators. The Ground truth for each image was created via majority vote. The test set consisted of 12 independent images. A validation set was only applied for the U-Net and consisted of another 15 Images. The U-Net was trained using PyTorch [28] in a Google Colab environment. To reduce the memory footprint of the model, images were split into 512x512 pixel-wide, not overlapping patches. The network was optimized with binary cross-entropy loss using stochastic gradient descent (SGD) with a learning rate of 0.01, momentum of 0.9 and a batch size of 8. Training continued until the loss in the validation dataset did not decrease further. In the presented case, this was true after 80 epochs.

With the applied QuPath network approach we exclusively utilised QuPaths built in functions. The annotation functions were used to segment and create the Ground-Truths that were used to train our networks. Instrumental for creation of our results were the training of a pixel classifier by artificial neural network (ANN_MLP), and the built in multifactorial cell counting function. Since we focused on the QuPath network, a complete workflow created exclusively with QuPath Version 0.4.3 is described in detail to ensure that other investigators can reproduce the approach to automate their own applications with the software. First, a project was created via the button "CREATE PROJECT". Afterwards, all recorded images of our samples were added to said Project with the "ADD IMAGES" command. After all images were added to the project, we created a neural network-based classifier to differentiate tissue area from background. To train a classifier, a certain amount of annotated image data with corresponding classes is needed. All annotation in this project was performed with QuPath's, built in annotation tools manually. In the "OBJECTS" window, each annotation was assigned the corresponding class of either "Tissue" or "Background". For more complex image data, more classes can be labelled, i.e. separate classes for different types of tissue or undesired artefacts that should be excluded from analysis. The classifier can utilize different parameters for its training, i.e. features that correspond to distinct visual or morphological appearance. Parameters should be chosen depending on the image data, they can be changed under "CLASSIFY → TRAIN PIXELCLASSIFIER → FEATURES → EDIT". We chose Gauss deviation, weighted deviation and gradient magnitude as tracked parameters. The classifier was trained with 20 fully annotated images for a maximum of 1000 iterations, before running over the whole test data. For every image in the project, the trained classifier decided if each distinct pixel is more likely to belong to the class "Tissue" or "Background". The live preview window should outline all pixels that are part of the tissue, otherwise adjustments to either the training data or parameters should be made. With "CREATE OBJECTS" annotations are created with the pixels that were previously classified by the neural network. To count all cells within an annotated region, based on pre-set values, we ran the command "ANALYSE → CELL DETECTION → CELL DETECTION" in the annotation region. For images of the same staining and tissue, most of these values should remain the same. The only value that can be changed through multiple iterations is the threshold value to determine the colour intensity at which a single cell should be detected. Since the same staining on another iteration might result in a different tissue colouration, adjusting this value slightly for each different staining is recommended. After counting the cells via this command, all measured values like detected cells or tissue area can easily be extracted for statistical analysis via the command "MEASURE → EXPORT MEASUREMENTS".

As performing all these steps for every image individually is time consuming, a script was created that ran these steps fully automated. QuPath utilises common programming language commands, therefore creating personalised scripts is simple. The command "AUTOMA-TE→SHOW WORKFLOW" opens the command history tab, displaying each previously

performed task. If the command history tab is opened, performing a new task with new set values automatically adds this new task to the list. With the button "CREATE SCRIPT" the software creates a script-based on all task entries in it. The script can be saved into the script folder, where it can be accessed for future projects by "AUTOMATE→SHARED SCRIPTS→YOUR SCRIPT". If a script is executed, a new window opens allowing to choose between "RUN" or "RUN FOR PROJECT". If "RUN FOR PROJECT" is selected, automated steps according to the script are performed for every image in the project.This workflow was used for analysis of tissue area detection of the automated methods.

To examine the reproducibility of the image analysis workflow, we created another full workflow with the QuPath network, following the above-mentioned protocol. This time to compare it to a manual approach with multiple investigators. Three different investigators annotated the tissue area of 31 test images manually, counting cell nuclei with ImageJ per hand. Afterwards, they each individually trained their own classifier with data from a manually annotated training-set of 14 pictures originating from three different bone samples. Then each individual classifier was used to annotate the same 31 image test-set of bone samples. In created annotations, cells were then counted with the built-in cell counting tool of QuPath. Test and training sets never contained the same images to avoid overestimation/overfitting of trained networks.

## Statistical analysis

Generated data were analysed using GraphPad PRISM Version 10 (GraphPad Software, Inc., San Diego, CA, USA). Normality of all data was tested using Shapiro-Wilk test. To compare the different segmentation methods, not normally distributed data like Jaccard indices were analysed with Kruskal-Wallis-test followed by Dunn correction for multiple testing. For comparison of the QuPath classifier with manual analysis a Wilcoxon matched pair signed rank test was used to analyse the difference between the coefficient of variances for tissue area and cells per tissue area, whereasdata was not normally distributed. A paired t-test was used to check for significant differences between the normally distributed data, like cell counts, area in pixel, area in $mm^2$ and counts per $mm^2$. For all statistical analyses, a value of $p < 0.05$ was considered significant (*$p \leq 0.05$; **$p \leq 0.01$; ****$p \leq 0.0001$). Further information is provided in the respective figure legends.

## Results

### Histological imaging of trabecular bone following cryosectioning

The protocol described above enables the generation of reproducible cryosections of trabecular bone tissue after decalcification. The sections are stainable with a variety of protocols, which allowed us to show staining functionality and tissue- as well as matrix-integrity (Fig 2). The structure of the cylindrical bone punches and platelets, including the physiological trabecular cavities, was preserved while staining was possible without any additional processing steps. To evaluate the quality of the created cryosections (Fig 2A) we compared them qualitatively to paraffin embedded sections (Fig 2B) from identical bone donors. Though no major differences could be observed, a slightly higher staining intensity was seen in the cryosections. Representative histological images of cryosections of HE-stained bone tissue with its spongy texture and blue stained cell nuclei are displayed in Fig 2C and 2D. To further prove functionality of multiple staining protocols, we performed fluorescence imaging (Fig 2E & 2F). Though the extracellular matrix showed high affinity for the DAPI (4′,6-diamidino-2-phenylindole) staining, a blue-fluorescent DNA stain that exhibits up to 20-fold enhancement of fluorescence upon binding to AT regions of dsDNA, cell nuclei are still highlighted and detectable (Fig 2E).

Fig 2F showed results of a TUNEL staining method to evaluate apoptosis in the tissue, we showed a DNase treated sample that served as a positive control. Apoptotic cells were labelled with TexasRed.

## Comparison of different automated tissue classification processes

We evaluated three different tissue detection methods regarding their ability to match a manually annotated image. Otsu thresholding, the built in QuPath pixel classification network and a convolutional neural network, U-Net (Fig 3). To compare the methods, we investigated the Jaccard index of each method, which represents the intersection over union between predicted masks to their corresponding ground truth. All three methods achieved high Jaccard Indices. In most cases the automatically segmented result replicated the manual data highly, differences

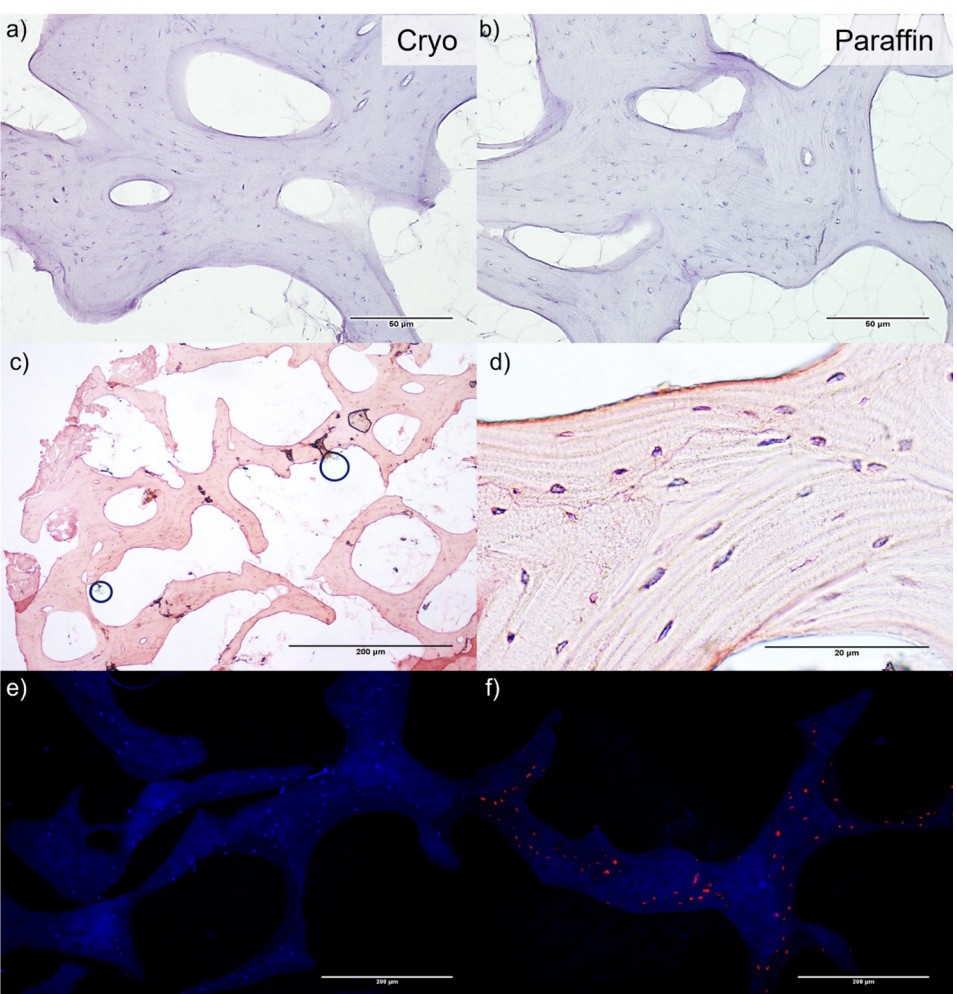

**Fig 2. Histological imaging of trabecular bone following cryosectioning.** Fig A) and B) represent a direct comparison of bone cryosections sections (A) and paraffin embedded sections (B). Both tissue sections were stained with Hematoxylin Eosin (HE) standard protocol. Scale bars represent 50 μm. No significant changes in quality and stainability of the tissues could be observed. Fig C) and D) show representative HE-stained bone tissue punches (with higher Eosin exposure captured with a light microscope at different magnifications, the pinkish red parts show bone tissue with its spongy texture, while the blue dots represent cell nuclei. Scale bars represent either C) 200 μm or D) 20 μm. Fig E) and F) show a TUNEL fluorescence staining of trabecular bone cryosections. Fig E) shows a negative control and Fig F) the positive control proving possible fluorescence staining after cryosection. Scale bars represent 200 μm.

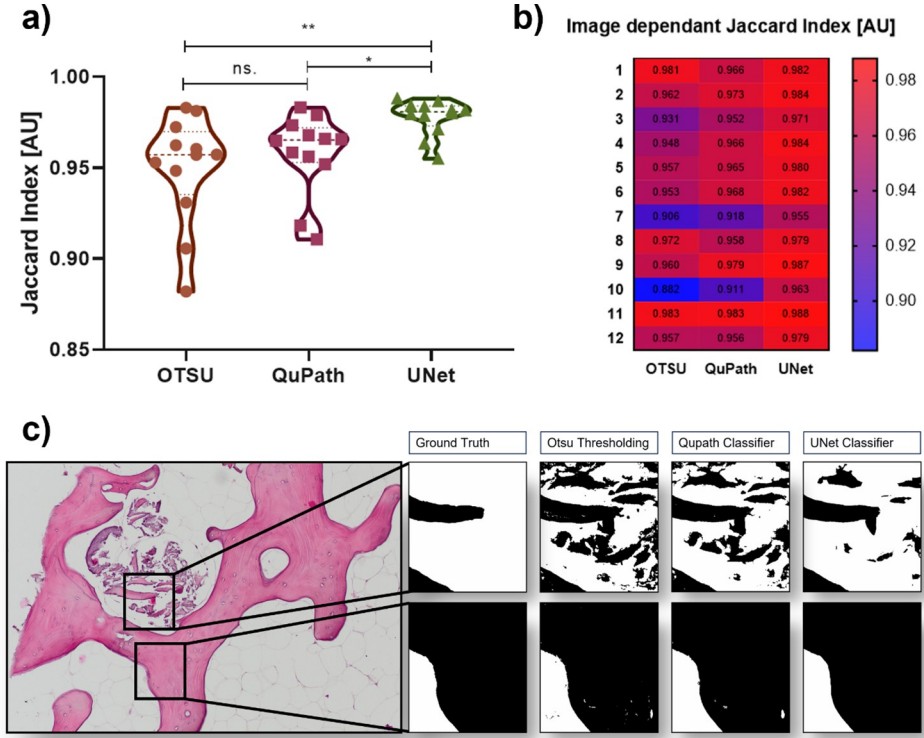

**Fig 3. Performance of different automated tissue area detection tools.** The figures highlight accuracy and performance of the three tested methods Ostu, Qupath and U-Net. A) shows the mean Jaccard indices of the three tools, U-Net was able to outperform the other methods significantly throughout the whole test set. To clarify and show image-dependent differences between the methods, Fig B) color codes and highlights the Jaccard indices for each image individually, ranging from the lowest intersection (blue) to highest (red). While in some images all methods performed nearly identical, other images resulted in image specific differences of their performance. To visualize classification patterns of the methods, we magnified areas of an exemplary test image C) that showed differences in performance. The upper lineup shows a highly difficult classification area where all methods struggled to match the ground truth, while the lower lineup shows a relatively well-matched area.

mostly stemming from tissue edge pixel- or small artifact-misclassification. The lowest mean value throughout all images was seen in the Otsu thresholding method (Mean = 94,9%±3,0%), surpassed by QuPath (Mean = 95.8%±2,2%) and U-Net (Mean = 97.8%±0,1%). Notably, the SD of the U-Net classifier was immensely reduced. A highly significant increase of accuracy could be observed between U-Net classification and the Otsu (adj.p = 0.004) thresholder as well as a significant increase of U-Net compared to the QuPath network (adj.p = 0,037). Between QuPath and Otsu no significant difference was observable (adj.p≥0,999) (Fig 3A). Segmentation performance was also observed for each image distinctively (Fig 3B). It was seen that while some images seem to produce similar results throughout all three testing methods, others seem to be problematic for thresholding via Otsu, resulting in relatively low intersection. Especially in these images, the network-based approaches created better fitting predictions. A complex picture "Image 10" was chosen from the test set to highlight the differences in classification patterns between the methods (Fig 3C).

## Reproducibility of manual and automated QuPath analysis

To evaluate the automated processing, inter investigator variance and biological relevance, three investigators each trained a neural network with the same training set. Test images were

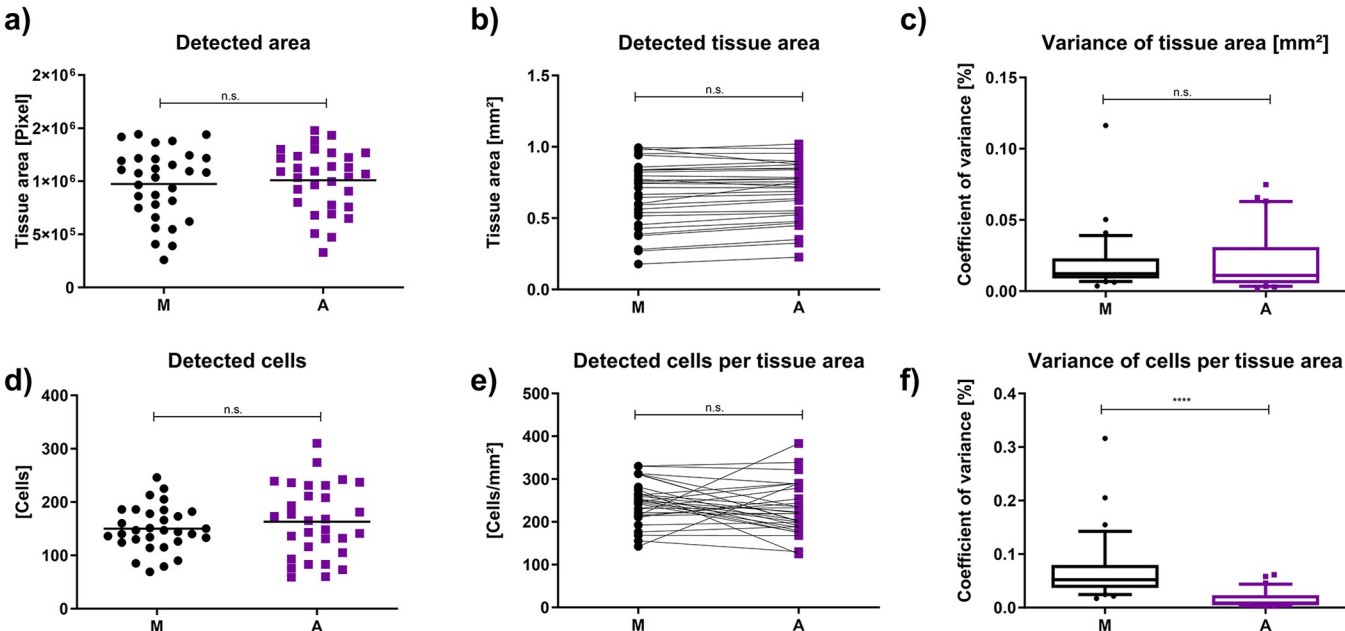

**Fig 4. Comparison of manual (M) and automated (A) analysis.** The figure presents manual data acquired by three different investigators (M) and compares it with three individually trained neural networks (A). Each dot represents the pooled result of manual investigation for a single image (n = 31), while squares represent the pooled result of the networks. Shown are differences in the detected pixel area (a) and the detected cells (d) as well as their derived biological weight (b and e). No significant differences could be observed in either of these data pools. Fig c) and f) visualize the accuracy of manual and automated analysis by comparing the coefficient of variance for either classified tissue area [mm$^2$] (c), or cell counts in detected tissue area (f). Data are given as median, whiskers as interquartile range.

then annotated, either manually with an ImageJ-based analysis or with a respective classifier, determining tissue area and counted cells in the detected area. Finally, the manually detected tissue area was compared with the automated tissue area regarding detected values, derived biological values and variance. (Fig 4).

The image data generated by either three manual investigators (M) or three automated scripts (A) were pooled for each image. The mean of manual investigators was compared with the mean of the automated analysis for each image.

We compared the calculated raw data for the detected area and cells between the two groups but found no significant difference regarding the classified pixels (P = 0.6655; Fig 4A) or detected cells (P = 0.3738; Fig 4D) showing that both manual and automated methods produce comparable results.

To show that not only the raw data but the derived biological results produced by the automated process match the manually determined cells and tissue areas, we investigated the detected tissue area and cells per mm$^2$ (detected cells/determined tissue area). The data demonstrates that results obtained with manually (M) and automated (A) processes show no significant differences in the detected tissue area (P = 0.6655; Fig 4B). Furthermore, it showed no significant difference in cells per tissue area (p = 0.1649; Fig 4E), highlighting that the automated process is able to reproduce the manually generated data.

To gain an insight into the differences between both approaches regarding accuracy, we evaluated the coefficient of variance. While the measurement of tissue area revealed no significant differences between the approaches (Fig 4C), we saw highly significant differences in numbers of cell count per tissue area (P<0.0001, Fig 4C).

## Discussion

In the present study, a technique for cryosectioning of bone tissue for histochemical analysis was established. It offers several advantages when compared to paraffin or epoxy resin embedded sectioning. The creation of histological specimens is fast and requires less technical steps and machinery, therefore offering a cheap and easy solution for research teams lacking the needed costly equipment. Created specimens with this cryosectioning method are not masked in any way, allowing staining with all common methods, e.g. fluorescence, HE or immunohistochemical staining. The resulting image quality for the analysis of bone hard tissue is comparable to the gold standard techniques. Unfortunately, during processing most of the bone marrow is washed out and analysis of it is therefore not achievable. To retain marrow structures, we recommend a shorter fixation in bousin solution, the creation of sections with 3–4μm thickness resulting in longer decalcification times and very gentle washing steps. Also, sections are relatively thick (8–10 μm) when compared to common approaches (3–5 μm) used for soft tissue cryosections [29, 30]. Nevertheless, the thickness of the slides does not interfere with stainability through standard protocols, but might complicate finding a focus level for microscopes or cameras. Because of this, cells may not be on the exact same focal plane. For analysis the investigator should choose a focus that will portray the slide representatively or, if the capturing setup allows it, consider techniques that allow for full image focus of thick samples like Z-Stack layering. While the method of choice for bone tissue will remain paraffin or other epoxy resin-based embedding and sectioning techniques, the presented method of cryosectioning offers a suitable alternative for laboratories with limited resources or technical devices to analyse bone specimen histologically. It is also of interest for labs struggling with staining of masked antigens.

Automated analysis offers great possibilities to handle large amounts of image data quantitatively in a fraction of the time manual analysis would require, while maintaining objective, investigator independent results. However automated processes are difficult to properly establish, especially for complex questions and difficult image data. It is also unable to notice spontaneous or unexpected findings, it is incapable of producing the same variability and modularity as a manual investigation would. For subsequent analysis, our aim was to create a fully automated protocol for trabecular bone image analysis that is adaptable for multiple research questions. We focused on establishing an adequate model for our data regarding computational complexity, accessibility and accuracy. Therefore, we chose three different approaches. Otsu thresholding was chosen as an easy and accessible automated thresholding method. It searches for a threshold that maximizes the mean difference between both classification classes in the histogram. This threshold is then applied to each pixel in the image, and used for classification. While Otsu thresholding serves as a reliable tool, pixels are not classified distinctively and without further context. More complex datasets and small differences in pixel intensity prove difficult to classify correctly with this method, as shown for a few images in our dataset (Image 3). While Otsu offers sufficient classification of the tissue area comparable to the QuPath network, we wanted to create a fully automated approach that allows for easy implementation of a subsequent cell detection, therefore we excluded Otsu thresholding. For bone tissue specifically, other possibilities for automated script-based analysis are already published, but they require extensive computational skills and knowledge of the MATLAB software [31]. Because of complex configuration details, classical script or algorithm-based workflows are hard to reproduce. They are heavily dependent on their configuration, and incorporate slight changes in their setup into their results. Neural network-based methods offer a great approach to achieve standardized and accurate results for quantitative analyses. They offer holistic results that are mostly independent of configuration. Therefore, achieved

results can be replicated by multiple investigators and labs, yet they are still rarely used in histologic analysis application fields. This limitation in application might be due to the challenginging code-heavy approach and the hard to access and even harder to navigate software that is necessary to utilize this approach. There is a variety of tools able to perform neural network-based analysis, but because of its easy possibility for automation and scaling, we chose QuPath. U-Net was chosen as an advanced convolutional network. Through its architecture it is able to learn complex features at multiple scales, incorporating spatial context, and leading to highly accurate predictions. With U-Net we trained a network that was able to outperform the other methods significantly (Otsu p = 0.0042; QuPath p = 0.0373). It gathers context information of each pixel within its receptive field, and is therefore able to generate accurate predictions of classification. However, it requires a lot of computational power and expertise to set up, and is therefore inaccessible for a large number of labs. While U-Net is a powerful tool to reliably classify even complex image data, the network trained with 'QuPath' offered sufficient results for our image data. The network was able to determine the tissue area based on their Jaccard index with a mean accuracy of roughly 95%±2% and showed no significant differences in the number of detected cells when compared to manual analysis. A noticeable advantage of 'QuPath' is it being an adaptive tool, with modular expansions, that still remains user-friendly due to a well-structured and easy to navigate user interface. Extraordinarily little IT knowledge is required to navigate the software efficiently and to understand how created scripts work as well as how to modify and use them. Original image data is not edited or manipulated in any way and remains the same. This offers easy data management, efficient storage capacity and extraction possibilities as well as even more built in features for analyses. This combined with it being a freely available software makes it accessible for nearly all laboratories. It is a continuously updated software, highlighting its future application safety. We evaluated the introduced QuPath protocol, by comparison to a conventional manual protocol, regarding its results. We were able to show that our generated raw data as well as the derived biological results from the automated process matched manually determined results. The detected tissue area, as well as the determined cell/area quotients from the network, showed no significant differences, which proves the credibility of performed automated analysis as a reliable alternative to manual detection. The variance of cell counts per tissue area was significantly lower in the automated approach than in the manual, this highlights the reliability of its built in multifactorial cell detection method to objectively detect cells. Cells are either detected or neglected objectively based on set parameters. Using the same parameters will always result in the same cell count, making it more robust and easier to reproduce than manual analysis. Even if the same image is analysed by the same investigator twice, slight differences in results are expected. Results from two different investigators may even vary heavily, as shown in other fields of image diagnostics and analysis [32, 33]. Interestingly, the variance in tissue detection shows no significant difference between human and classifier tissue detection, even though one would predict a lower variance in the trained classifiers as well. The received training is most likely responsible. The networks were trained by different investigators, each implementing slight differences regarding measurements in its training, consequently the classifier replicates differences and adjusts its predictions slightly. In practice, a lower variance of a trained classifier is still predicted. A given investigator can use their formerly created classifier and script to create an exact copy of the results with the same images, while a manual approach would create minor differences resulting in an iteration-dependent variance that is absent in an automated process. The presented method can handle large sets of images with the same accuracy and results as a manual approach in less time. With larger datasets, it is just as accurate as manual counting techniques, while maintaining a significantly lower inter-investigator variance in cell counting compared to a manual approach (P<0.0001).

Although we used this analysis protocol for bone tissue specifically, it works for a wide variety of different tissues and scientific questions. Presented method can easily be used to segment a number of different tissues from their respective background, it could even be further enhanced to separate and identify multiple tissue types from another. Other researchers willing to establish a similar method for their own respective research interest should consider that network-based approaches need some setup time. Certain parameters need to be adjusted depending on the image data provided. The number of training samples needed to create an accurate classifier for other applications varies heavily with the complexity of classification and image data. With more data to learn from, a neural network gets more potent at the classification process, so increasing sample numbers will make the classifier more reliable up to a certain degree. If correctly trained, neural network-based classifiers can reproduce expert analyst's classification as previously shown with radiological imaging [34]. However, with the possibility of automation network-based methods like the one presented especially save a lot of time when analysing large sets of images, allowing for analysis otherwise too burdensome or simply unachievable by manual means, because of the otherwise extraordinary workload.

## Conclusion

We were able to create a reliable cryosection protocol for trabecular bone tissue sections, and combined it with a functional fully automated image analysis approach. When compared with other automated methods as well as with a manual approach, it showed satisfying results that matched the manual gold standard, therefore creating a reliable alternative. Though manual analysis methods will remain the gold standard in quantitative histology for now, the possibility to change is given with innovative tools and easily accessible and navigable neural networking software. To validate histology for quantitative analysis, it needs to be able to objectively quantify large datasets. In the past this was always a limitation to classical histology because of the connected workload. The need for quantifiable results from actual imaging of tissue and cells alike is still rising and deep analysis-based protocols offer solutions to connected problems. Hence, we are positive that network-based analysis will gain more importance with time, based on its flexibility in use, its ability to handle large amounts of data in minimal time, as well as its reliability and will therefore continue to find its way into quantitative histology and even more scientific fields.

## Acknowledgments

We thank Kirsten Kesselring, Antje Janetzko (Department of General Surgery, Visceral, Thoracic and Vascular Surgery, University Medical Center Greifswald) and Daniel Wolter (Department of Oral, Maxillofacial and Plastic Surgery, University Medical Center Rostock) for their excellent technical support.We also want to express our thanks to the Institute of Anatomy, Rostock University Medical Centre for providing us with the bone samples used in this publication.

Image 1 was created with BioRender.com.

## Author Contributions

**Conceptualization:** Christopher Pohl.

**Data curation:** Janine Waletzko-Hellwig.

**Funding acquisition:** Rainer Bader, Stephan Kersting.

**Investigation:** Nico Brandt, Charlotte Koppe.

**Methodology:** Nico Brandt.

**Project administration:** Andreas Hoene.

**Resources:** Janine Waletzko-Hellwig.

**Software:** Moritz Kunzmann.

**Supervision:** Rainer Bader, Michael Schlosser, Andreas Hoene.

**Validation:** Janine Waletzko-Hellwig, Friederike Kalle.

**Visualization:** Christopher Pohl, Moritz Kunzmann.

**Writing – original draft:** Christopher Pohl.

**Writing – review & editing:** Moritz Kunzmann, Rainer Bader, Stephan Kersting, Daniel Behrendt, Andreas Hoene.

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
