## [Decision Letter · Decision Letter 0]

7 Dec 2023

PONE-D-23-31772Quantitative analysis of trabecular bone tissue cryosections via a fully automated neural network-based approachPLOS ONE

Dear Dr. Pohl,

Thank you for submitting your manuscript to PLOS ONE. After careful consideration, we feel that it has merit but does not fully meet PLOS ONE’s publication criteria as it currently stands. Therefore, we invite you to submit a revised version of the manuscript that addresses the points raised during the review process.

We look forward to receiving your revised manuscript.

Kind regards,

Syed M. Faisal, Ph.D.

Academic Editor

PLOS ONE

Journal Requirements:

 "This work was funded by the ESRE, ESF and ELER as well as the Ministry of education, science, and culture of Mecklenburg Western Pomerania ‘Exzellenzforschungsprogramm’ and is part of the HOGEMA joint research project under the sponsorship contract ESF/14-BM-A55-0015/18."

"This work was funded by the ESRE, ESF and ELER as well as the Ministry of education, science, and culture of Mecklenburg Western Pomerania ‘Exzellenzforschungsprogramm’ and is part of the HOGEMA joint research project under the sponsorship contract ESF/14-BM-A55-0015/18.

We thank Kirsten Kesselring, Antje Janetzko (Department of General Surgery, Visceral, Thoracic and Vascular Surgery, University Medical Center Greifswald) and Daniel Wolter (Department of Oral, Maxillofacial and Plastic Surgery, University Medical Center Rostock) for their excellent technical support.

We also want to express our thanks to the Institute of Anatomy, Rostock University Medical Centre for providing us with the bone samples used in this publication.

Image 1 was created with BioRender.com."

"This work was funded by the ESRE, ESF and ELER as well as the Ministry of education, science, and culture of Mecklenburg Western Pomerania ‘Exzellenzforschungsprogramm’ and is part of the HOGEMA joint research project under the sponsorship contract ESF/14-BM-A55-0015/18."

6. Please amend your authorship list in your manuscript file to include author Dr. Michael Schlosser.

Reviewers' comments:

Reviewer's Responses to Questions

**Comments to the Author**

1. Is the manuscript technically sound, and do the data support the conclusions?

Reviewer #1: Yes

Reviewer #2: Yes

2. Has the statistical analysis been performed appropriately and rigorously? 

Reviewer #1: Yes

Reviewer #2: Yes

3. Have the authors made all data underlying the findings in their manuscript fully available?

Reviewer #1: Yes

Reviewer #2: Yes

4. Is the manuscript presented in an intelligible fashion and written in standard English?

Reviewer #1: Yes

Reviewer #2: Yes

5. Review Comments to the Author

Reviewer #1: In this article, authors developed a fully automated technique for cryosectioning of bone tissue for histochemical analysis. It showed many advantages when compared to paraffin or epoxy resin embedded sectioning and creation of histological specimens is fast and requires less technical steps and machinery. Therefore, it offers a cheap and easy solution for research teams lacking the needed costly equipment. The manuscript is well written and the experiments are well-designed. I have only a few concerns about the manuscript.

-In discussion section, it is mentioned that during the processing bone marrow was washed away therefore not analyzed, so can you suggest any modification in the protocol followed by you, so that bone marrow could also be imagined?

-In line 111, include period before “Lately”

-In line 114, remove period after “analysis”

-In line 116, remove network after “neural”

-In line 155, replace “D” with “d” in Decalcification

Reviewer #2: The article introduces a novel cryosectioning protocol for trabecular bone tissue and combines it with automated image analysis techniques. This protocol demonstrated reliability in comparison to manual methods, showcasing its potential as an alternative approach. While manual analysis remains the gold standard in histology, the study suggests that easily accessible neural networking software and automated methods hold promise due to their flexibility, efficiency in handling large datasets, and reliability, potentially shaping the future of quantitative histology and other scientific fields.

Methodology and Protocols:

Could you provide further details on the specific steps involved in the cryosection protocol developed for trabecular bone tissue?

Comparative Analysis:

What were the specific strengths and limitations observed in the comparison of automated image analysis methods versus manual analysis?

Software and Tools:

How was the QuPath software utilized in the image analysis process, and what specific features or tools within QuPath were instrumental in achieving the study's objectives?

Were there any challenges or limitations encountered while implementing the neural network-based approach using UNet? How were these challenges addressed?

Results and Validation:

Can you provide more details on the statistical methods used to validate the automated image analysis results against manual analysis? Were there any statistical tests or measures of agreement used?

In what specific aspects did the automated analysis show variations or discrepancies compared to manual analysis, if any?

Limitations and Future Directions:

What future applications or enhancements do you envision for this methodology, beyond trabecular bone tissue analysis, and how might these be addressed?

Conclusion and Implications:

Can you expand upon the implications of the study's findings for the field of histological analysis and its potential impact on research practices?

How might the study's results influence the adoption of automated image analysis methods in histology, considering the current reliance on manual analysis?

6. PLOS authors have the option to publish the peer review history of their article (what does this mean?). If published, this will include your full peer review and any attached files.

Reviewer #1: No

Reviewer #2: No

---

## [Author Response · Author response to Decision Letter 0]

24 Jan 2024

All responses to the specific reviewers and editor comments are compiled in the attached document Response to the Reviewers. We hope we were able to respond to all arising questions and remarks sufficiently.

---

## [Editor Report · Decision Letter 1]

31 Jan 2024

Quantitative analysis of trabecular bone tissue cryosections via a fully automated neural network-based approach

PONE-D-23-31772R1

Dear Dr. Pohl,

We’re pleased to inform you that your manuscript has been judged scientifically suitable for publication and will be formally accepted for publication once it meets all outstanding technical requirements.

Kind regards,

Syed M. Faisal, Ph.D.

Academic Editor

PLOS ONE
---

## [Editor Report · Acceptance letter]

8 Feb 2024

PONE-D-23-31772R1 

PLOS ONE

Dear Dr. Pohl, 

I'm pleased to inform you that your manuscript has been deemed suitable for publication in PLOS ONE. Congratulations! Your manuscript is now being handed over to our production team.

Kind regards, 

on behalf of

Dr. Syed M. Faisal 

Academic Editor

PLOS ONE